# Delivering effective, comprehensive, multi-exercise component cardiac rehabilitation (CR) for chronic heart failure patients in low resource settings in sub-Saharan Africa: Queen Elizabeth Central Hospital—(QECH-CR) randomised CR study, Malawi

**Alice Namanja** [1]*, **Daston Nyondo**[2], **Tendai Banda**[2], **Ephraim Mndinda**[2], **Adrian Midgely** [3], **James Hobkirk**[4], **Sean Carroll**[4], **Johnstone Kumwenda**[5]

**1** Rehabilitation Sciences Department, Kamuzu University of Health Sciences, Blantyre Malawi,
**2** Physiotherapy Department, Queen Elizabeth Central Hospital, Blantyre Malawi, **3** Sport and Physical Activity, Edge Hill University, England, United Kingdom, **4** School of Sport, Exercise & Rehabilitation Sciences, University of Hull, England, United Kingdom, **5** School of Medicine and Oral Health, Kamuzu University of Health Sciences, Blantyre Malawi

* anamanja@medcol.mw

## Abstract

### Background

The delivery of Cardiac Rehabilitation (CR) and attaining evidence-based treatment goals are challenging in developing countries, such as Malawi. The aims of this study were to (i) assess the effects of exercise training/ CR programme on cardiorespiratory and functional capacity of patients with chronic heart failure (CHF), and (ii) examine the effectiveness of a novel, hybrid CR delivery using integrated supervised hospital- and home-based caregiver approaches.

### Methods

A pre-registered (UMIN000045380), randomised controlled trial of CR exercise therapy in patients with CHF was conducted between September 2021 and May 2022. Sixty CHF participants were randomly assigned into a parallel design-exercise therapy (ET) (n = 30) or standard of care (n = 30) groups. Resting hemodynamics, oxygen saturation, distance walked in six-minutes (6MWD) and estimated peak oxygen consumption (VO$_2$ peak) constituted the outcome measures. The exercise group received supervised, group, circuit-based ET once weekly within the hospital setting and prescribed home-based exercise twice weekly for 12 weeks. Participants in both arms received a group-based, health behaviour change targeted education (usual care) at baseline, 8-, 12- and 16-weeks.

### Results

Most of the participants were female (57%) with a mean age of 51.9 ±15.7 years. Sixty-five percent (65%) were in New York Heart Association class III, mostly with preserved left

**Data Availability Statement:** All relevant data are within the manuscript and its Supporting information files.

**Funding:** The first author (Alice Namanja) received a small research grant from the Malawi NCD BRITE consortium to implement this project. The funders also offered research writing skill development and implementation science. NCD BRITE Consortium assigned Alice Namanja to a local expert in the field of Cardiovascular disease prevention and management. Together, with other identified collaborators, they developed the project, implemented it and prepared the manuscript. The funders had no role in study design, data collection and analysis, decision to publish, or preparation of the manuscript.

**Competing interests:** The authors have declared that no competing interests exist.

ventricular ejection fraction (HFpEF) (mean Left Ventricular Ejection Fraction 52.9 ±10.6%). The 12-weeks ET led to significant reductions in resting haemodynamic measures (all P <0.05). The ET showed significantly higher improvements in the 6MWD (103.6 versus 13.9 m, p<0.001) and $VO_2$ peak (3.0 versus 0.4 ml·kg$^{-1}$·min$^{-1}$, p <0.001). Significant improvements in 6MWD and $VO_2$ peak (both p<0.001), in favour of ET, were also observed across all follow-up timepoints.

## Conclusion

This novel, randomised, hybrid ET-based CR, delivered to mainly HFpEF patients using an integrated hospital- and home-based approach effectively improved exercise tolerance, cardiorespiratory fitness capacities and reduced perceived exertion in a resource-limited setting.

## Introduction

Chronic Heart Failure (CHF) often constitutes a final pathway for chronic cardiovascular diseases (CVD), which constitute the leading cause of mortality and morbidity worldwide [1, 2] with low- and middle-income countries accounting for >80% of the CVD global burden [3]. Sub-Sahara Africa (SSA) experiences nearly a 2-fold high annual rate of CHF-related mortality (35%) compared to other parts of the world (16.5%) [4, 5], and it typically involves adults within the economically active age groups [6]. In 2016, CVD attributed to 10% of all deaths in Malawi [7]. The growing burden of CVD in SSA is attributed to the rising burden of obesity, physical inactivity, heavy alcohol use, Type 2 diabetes, hypertension, and smoking prevalence [8]. A high prevalence of such cardiac risk factors has also been observed among people in Malawi [9, 10].

CHF is typically characterised by symptoms such as shortness of breath, fatigue, and reduced exercise tolerance. CHF symptoms account for more than 30% of hospitalizations in Africa [11] and present a considerable burden on the developing healthcare delivery systems in the region.

The goals of treatment are to slow the progression of the CVD, improve patients' quality of life (QoL) and their ability to function physically, by managing the clinical syndrome symptoms. Cardiac Rehabilitation (CR) is a comprehensive management strategy aimed at promoting patients' physical, mental, and social well-being through exercise therapy (ET), lifestyle-change targeted education and counselling, and monitoring and enhancement of adherence to medications [12]. Comprehensive evidence from the developed countries demonstrate that exercise-based CR positively reduces CHF-related hospitalizations by almost 30–42% and improves health related QoL [13, 14]. Exercise training, especially higher intensity intermittent exercise, is now widely recommended for CHF patients due to positive improvements in functional class, functional capacity and QoL outcomes [15]. CR also significantly increases exercise tolerance capacity, and directly determined peak oxygen consumption ($VO_2$peak) (by 3.8 ml·kg$^{-1}$·min$^{-1}$) among patients with CHF [16]-although research has been conducted largely on CHF participants with reduced LV function.

Heart failure patients with preserved ejection fraction (HFpEF) also experience reduced exercise capacity, which is associated with a poor prognosis and impaired QoL [17]. Exercise training was shown to improve functional capacity and physical dimensions of QoL within an

early randomized controlled trial (RCT) in older, symptomatic patients with preserved systolic function (New York Heart Association classes II and III) [17]. Peak $VO_2$ and six-minutes' walk distance (6MWD) increased by 3.3 ml·kg$^{-1}$·min$^{-1}$ and 6.0 m more following ET, respectively, and remained unchanged within the usual care intervention. The improvement in peak $VO_2$ corresponded to a 24m increase in the 6MWD in patients who participated in the exercise training programme and no significant changes were evident in the usual care group. The positive change in $VO_2$peak was associated with atrial reverse remodelling and improved left ventricular diastolic function [17]. Subsequently, within a pooled data analysis of 6 RCTs, patients with HFpEF undergoing exercise training had significantly improved cardiorespiratory fitness (CRF) (weighted mean difference, 2.7 ml·kg$^{-1}$·min$^{-1}$; 95% confidence interval, 1.8–3.7 ml·kg$^{-1}$·min$^{-1}$) and quality of life outcome when compared with the control group [18]. Further, systematic reviews by Long *et al.* [19] and Dibben *et al.* [14] suggest that CR may reduce short-term (up to 12 months) risk of all-cause and heart failure specific hospitalizations. It has been recommended that future exercise training within CR trials should consider recruiting traditionally less represented CHF patient groups including older, female, and HFpEF patients, and use alternative CR delivery settings including home- and technology-based programmes [13].

In this regard, CR is underutilized on the African continent and has been reported to be non-existent in Malawi and many other parts of Africa [20], and patients with CHF rarely receive the holistic CR intervention. Several models of delivering CR have been suggested, which encourage developing settings to use low-cost resources and approaches to increase accessibility and utilization of the intervention [21, 22]. It is not known if CR could effectively be delivered in Malawi, using proposed CR delivery models. Therefore, this study aimed to assess effects of CR on exercise tolerance, symptoms and other CRF parameters amongst people living with CHF. The exercise training intervention utilised in the Queen Elizabeth Central Hospital CR (QECH-CR) study was distinct in that it incorporated both an intermittent, circuit-based exercise protocol of moderate to vigorous aerobic and resisted exercise; and an integration of supervised and home-based training. The study also aimed to determine underlying factors that may act as barriers to delivering and utilising the intervention within this setting.

## Materials and methods

### Design, population, and eligibility criteria

A prospective clinical trial was conducted at QECH, Physiotherapy department, Blantyre, Malawi, between September 2021 to May 2022. The protocol was registered in the University Hospital Medical Information Network Clinical Trials Registry (UMIN-CTR) (UMIN000045380). Patients with CHF were enrolled from the outpatient chest clinic of the facility if they were aged 18 years and above, had a cardiac echocardiogram undertaken and had provided informed consent for a randomised parallel-design exercise-based CR or usual care. They were excluded if they had any known contraindication to performing physical activity (PA), were classified as being at a high risk for cardiac events during exertion using the British Association for Cardiovascular Prevention and Rehabilitation (BACPR) risk stratification guidelines [12], were not willing to commit to subsequent and frequent follow-up visits and did not consent to participation. Both verbal and written consents were sought from the participants. Each participant was briefed verbally about the study by the research team, and they were given a detailed information sheet to read and understand before deciding on participation. Thereafter, informed consent was obtained from the participants who expressed interest to participation, by signing the consent form in the presence of a research team member.

## Sample size and sampling technique

The sample size for this study was calculated to detect a mean difference of 0.6 ml.kg$^{-1}$.min$^{-1}$ in the Pred.VO$_2$peak [23] between the groups, at 80% power with a two-tailed significance level set at 0.05. A total of 80 participants, 40 in each arm (1:1), was required. Due to COVID-19 restrictions in the operations of the chest clinic, patient turnover and recruitment was remarkably reduced during the originally proposed study period. The study was subsequently extended by four months to account for these circumstances. The study team managed to contact 73 patients who met the study inclusion criteria and enrolled 60 CHF participants (25% deficit of the actual calculated sample size) over the extended study period. To minimize selection bias, all patients were selected from the sequentially scheduled routine chest clinic appointments at the hospital. Patients were randomly allocated to a parallel-design ET intervention or usual-care group using selection from sealed envelopes by a neutral person. Numbers 1 and 2 were each wrote on a piece of paper, folded, mixed, and placed in sealed envelopes. The numbers were pre-defined prior to randomization, by the primary investigator, by what intervention each represented. Fig 1 is a consort-type flow chart describing the selection of participants and sample size at each phase of the study. The start of data collection was delayed due to procurement and resource administration and management hiccups from the funders' side, which were beyond researchers' control. Initially, two months was assigned for recruitment, and it was extended by one more month due to low turnover of patients at the clinic. Recruitment started on 13$^{th}$ September 2021 and ended on 22$^{nd}$ November 2021 (due to routine end of year closure of the Chest clinic). Participants were identified and approached, in the Chest clinic, by the researchers. Table 1 describes the general baseline characteristics of the participants enrolled in the study.

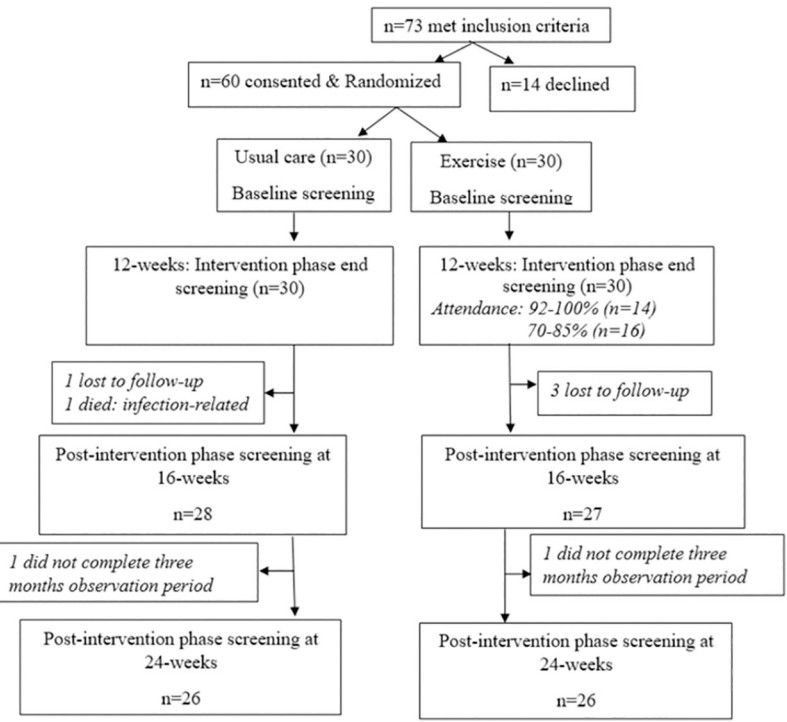

**Fig 1. A flow chart of sample size selection and availability at each study's screening point.**

**Table 1. Baseline characteristics of participants recruited in this study.**

| Intervention | Mean Age (±SD) years | Gender (n) (Male/female) | Follow-up period 12-weeks (participant n = 60) | Follow-up period 16-weeks (participant n = 55) | Follow-up period 24-weeks (participant n = 52) |
|---|---|---|---|---|---|
| Usual care | 53.1(17.3) | 10/20 | 30 | 28 | 26 |
| QECH Exercise therapy | 50.8(14.2) | 16/14 | 30 | 27 | 26 |

## Screening procedure

All participants underwent assessments at baseline, and after 8–12-, 16- and 24-weeks. The baseline evaluation involved collecting sociodemographic and medical history data from each participant, and measuring anthropometric, hemodynamic, and functional capacity tests. The medical history (r)elated data such as left ventricular ejection fraction (EF), list of medications and comorbidities were collected subjectively from participants and(or) verified retrospectively using medical records documented in their personal health passport booklet (in custody of the participants). The primary outcome measures included 6-Minute Walking Distance (6MWD) from submaximal walk test and estimated $VO_2$ peak ($VO_2$peak); and secondary measures included Body Mass Index (BMI), Systolic blood pressure (SBP), Diastolic blood pressure (DBP), Heart rate (HR) and Oxygen saturation ($O_2$Sats), respectively. Participants in the treatment group additionally underwent the Chester Step Test for cardiorespiratory fitness estimation and exercise prescription purposes at baseline and follow-up. The anthropometry and cardiorespiratory fitness measures, and participants' barriers to engaging in physical exercise at home were recorded at the subsequent follow-up time points. On each assessment timepoint, all participant in both study arms were asked to report if they faced any challenge which hindered them from engaging in PA at home, and the challenges were recorded on a checklist. At week 12, the physiotherapists involved in the delivery of the programme also discussed the challenges encountered in delivering the hospital-based supervised group-based intervention to the participants, which may hinder accessibility and utilization of the intervention at the facility and they recorded observations on a checklist.

To calculate BMI ($kg/m^2$), participants' body mass (kg) and stature (m) were first measured using an integrated stadiometer and digital weighing scale device (Physician scale 500lB, Henry Schein Inc.). Resting sitting blood pressure (mmHg) was measured manually using ADC Aneroid manual sphygmomanometer and $O_2$Sats (%) and HR (beats/minute) were recorded from a finger-probe pulse oximeter (MD300 C29, ChoiceMMed).

To assess functional capacity, six-minute walk test (6MWT) was conducted. The test is a valid and reliable indicator of functional capacity in patients with CHF, a strong prognostic marker of the severity of CVD, and it is related to both $Vo_2$peak and clinical (NYHA) functional classification [24]. In patients with advanced CHF evaluated for cardiac transplantation, distance ambulated during the 6MWT (mean 310±100m) strongly predicted $VO_2$peak and short-term event-free survival [25]. Participants were asked to walk a 20m long corridor, to and back around two cones placed at each end, for a period of 6-minutes. The 20m long corridor was used because this was the only available space at the facility, which could be used for patient assessment without interruptions. The walking was self-paced, but it was clearly explained to the participants that the aim of the test was to see how much distance they can manage to cover in 6-minutes. Participants were allowed to rest during the walk to catch breath if they needed it and were encouraged to continue walking as soon as they felt better. Every minute HR and $O_2$Sats were recorded using the oximeter. The $VO_2$peak was predicted using Cahalin et al. [25] equation ($VO_2$ peak = 0.03*distance(m)+3.98). Mean $VO_2$peak

difference values (0.1 ml·kg$^{-1}$·min$^{-1}$), coefficient of determination ($R^2 = 0.97$), and mean absolute percentage error (0.14) compared to criterion, ten-stage treadmill ramp protocol with direct respiratory gas analysis indicates that the Cahalin et al. equation provided the strong and best predictive validity compared to other existing 6MWT equations [26]. All participants performed the 6MWT at baseline, 8-, 12-, 16- and 24 weeks. On each day of testing, each participant had one session of 6MWT.

Chester step test, which is also a reliable test in the estimation of submaximal cardiovascular responses in patients with CVD [27], was conducted on participants in the treatment arm for exercise prescription. Participants were asked to step on either 6' or 8' inch box, onto and down repeatedly, following metronome beats played on a cassette. The test has 5 stages, each lasting 2-minutes; speed in the first stage is 15steps.minute$^{-1}$ and increases by 5steps.min$^{-1}$ at each stage. HR and Rating of Perceived Exertion (RPE) were recorded at each stage, using finger-probe oximeter and Borg-20 RPE chart respectively. One repetition maximum (1RM) was estimated in sitting position, with participant's back rested on the backrest of the chair, knees, and hip at the right angle, using free weights in form of weighted sand bags. Participants selected their perceived highest load they can carry in one load with their maximal effort and were guided through the movement of the limb to prevent compensatory motions. Whilst in sitting position, the researcher handed over the load in the participant's hand, and they were advised to lift it up to their chest by moving the elbow joint (flexion and extension) without moving the shoulder joint. Participants who were able to lift the same load up for the second time were asked to rest for a minimum of 15 minutes and were allowed to select another increased load to repeat the test until the 1RM was identified.

## Intervention and implementation procedure

Participants in the treatment arm performed both supervised aerobic and resistance exercise training, 3-times, and 2-times a week respectively for a 12-week period. Participants undertook one supervised session at the hospital in the Physiotherapy department, and two supervised sessions at home every week. Each participant within the CR intervention group was given a stepping box and were taught to source the traditionally made Thera-bands and make improvised weights using sand in plastic bottles, or clothes for home-based exercise. Caregivers were advised to weigh the improvised weights to ensure it is of the prescribed size for the participant. At baseline, participants in both study groups came with their caregivers, who were trained on how to supervise and encourage the participants to do PA within home settings. The hospital-based ET protocol involved treadmill walking, box stepping, marching, ball throwing, cycling, weight lifts and TheraBand stretches. The prescribed home-based ET involved flat ground walking, marching, box stepping, weightlifts and TheraBand stretches. Aerobic exercise was prescribed at 40–70% of heart rate reserve (or 12–14 RPE) based on age and medication-related estimation of maximal heart rate. Muscular strength exercises were prescribed at 30–40% 1 repetition maximum for upper extremities and 50–60% 1 repetition maximum for lower body exercises. The strength exercises were performed at incrementally 2–4 sets of 10 repetitions, while the continuous aerobic training was undertaken for 30 minutes on each session. Circuit interval training, using a group approach, was employed at the hospital. The ET at the hospital and home begun with warm up and ended with cool down, each phase lasting 15 minutes and 10 minutes respectively. The warm-up and cool-down phases involved light stretches, spot marching and dancing steps.

As a standard care, all participants in both arms received a group lifestyle change targeted education and counselling. Participants and researchers discussed the risk factors for CVD progression and prevention measures. Participants were all encouraged to engage in PA at

home and take rest in between if they feel out of breath or experience symptoms such as angina, dizziness, shortness of breath and fatigue. The group-based lifestyle change targeted education and counselling was delivered at baseline, 8-, 12- and 16-weeks. No hospital-based ET was delivered after week 12; participants were advised to continue with self-care at home under the supervision of their caregivers. The intervention begun on the 15th September, 2021 and ended on 23rd February 2022.

## Data analysis

IBM SPSS version 28.0 was used to analyse the data, and results are mainly presented as mean ±SD or n (%). To compare the mean differences in all variables between participants in the ET and usual care arms at baseline, Univariate analysis of variance (Anova) analyses were conducted. To compare the mean change differences between the groups at the end of 12-weeks supervised ET period, the mean differences in each variable were calculated for each study group between the data collected at baseline and 12-weeks. This approach was considered as appropriate as there were not participant dropouts at this stage of the study. As a sub-group analysis, paired t-tests were conducted to assess the changes in HR and RPE responses to exercise, among participants in the ET group, at each stage of Chester step test. Analysis of Covariance Variance analyses (with age, sex and HIV diagnosis as covariates) were also conducted on baseline to 12-week follow-up and the statistical significance level was set at P-value <0.05.

Mixed model analyses were conducted to examine the group (exercise intervention versus usual care) responses over time in the primary (6MWD and $VO_2$peak) and secondary (DBP, SBP, O2Sats, BMI and HR) variables across the five assessment points (undertaken at baseline, eight-, 12-, 16-, and 24-weeks time-points). Different covariance structures, which were appropriate for data with unequal time-points between repeated measures, were evaluated. The data for the unstructured covariance matrices are present -which was typically the best fitting covariance structure that minimised the Akaike's information criterion [28]. Estimates of type III tests of fixed effects were reported for groups (treatment versus control) time and interaction effects. Estimated marginal means were evaluated and adjustments for multiple comparison undertaken with the Sidak test. Statistical significance was set at P value < 0.05.

## Ethical and safety considerations

The study was reviewed and approved by the Kamuzu University of Health Sciences research ethics committee (P.10/20/3167), and informed consent was obtained from each participant. All participants were screened for risk of cardiac events, and vital signs were monitored before and after the exercise sessions. Caregivers were trained to supervise PA at home, and the hospital-based exercise was supervised by physiotherapists.

## Results

### Socio-demographic characteristics of participants

Majority of participants recruited to the study were relatively young (51.9±15.7 years), mostly females (56.7%), married (78%), and with a history of hypertension (83%). Most were diagnosed with CVD within the last 5 years and a high proportion of participants were on relevant optimal CHF therapies, including diuretics (92%), beta blockers (62%) and Angiotensin Converting enzymes (55%) as shown in Table 2. Most participants did not engage in routine prior PA or structured exercise programmes (85%). Baseline echocardiography showed that left ventricular EF of the enrolled participants was mostly preserved or mid-range with a mean of 52.9 (±10.6%); approximately 22% of participants had evidence of reduced LV systolic function.

**Table 2. Comparison of socio-demographic characteristics of participants between the groups.**

| | Usual care (n = 30) | Exercise therapy (n = 30) | P-value |
|---|---|---|---|
| **Mean age years (SD)** | 53.1(±17.3) | 50.8(±14.2) | 0.49 |
| **Gender n(%)** | | | |
| Male | 10(33.3) | 16(53.3) | 0.19 |
| Female | 20(66.7) | 14(46.7) | |
| **Marital status n(%)** | | | |
| Single | 2(6.7) | 5(16.7) | |
| Married | 24(80.0) | 23(76.7) | 0.87 |
| Divorced | 1(3.3) | 0(0) | |
| Widowed | 3(10) | 2(6.7) | |
| **CVD duration n(%)** | | | |
| ≥ 10 years | 2(6.7) | 3(10.0) | |
| 6–9 years | 5(16.7) | 1(3.37) | 0.40 |
| ≤5 years | 23(76.7) | 26(86.7) | |
| **Hypertensive n(%)** | 23(76.7) | 27(90.0) | 0.17 |
| **Diabetic n(%)** | 4(13.3) | 7(23.3) | 0.32 |
| **HIV+ n(%)** | 12(40.0) | 15(50.0) | 0.44 |
| **Asthmatic/COPD n(%)** | 3(10.0) | 1(3.3) | 0.61 |
| **Smoking n(%)** | 0(0.0) | 1(3.3) | 0.50 |
| **Alcohol n(%)** | 1(3.3) | 4(13.3) | 0.35 |
| **PA engagement n(%)** | 3(10.0) | 6(20.0) | 0.47 |
| **Mean EF (SD)** | 51.5(±10.4) | 54.4(±10.8) | 0.30 |
| **HF Classes** | | | |
| HFrEF | 7(23.3) | 6(20.0) | - |
| HFmrEF | 8(26.7) | 6(20.0) | |
| HFpEF | 15(50.0) | 18(60.0) | |
| **NYHA Classes** | | | - |
| II | 11(36.7) | 10(33.3) | |
| III | 19(63.3) | 20(66.7) | |
| **Medications n(%)** | | | |
| Beta blocker | 17(56.7) | 20(66.7) | |
| ACE | 19(63.3) | 14(46.7) | |
| ARB | 1(3.3) | 1(3.3) | |
| Diuretics | 27(90.0) | 28(93.3) | |
| CCB | 9(30.0) | 10(33.3) | |
| Digoxin | 3(10.0) | 0(0) | - |
| Isosorbide | 0(0) | 1(3.3) | |
| Metformin | 1(3.3) | 5(16.7) | |
| Glibenclamide | 1(3.3) | 3(10.0) | |
| Bronchodilators | 0(0) | 2(6.7) | |
| Ferrous sulphate | 1(3.3) | 0(0) | |
| Aspirin | 5(16.7) | 8(26.7) | |

*HIV-Human Immunodeficiency Virus; EF-Ejection fraction; SD-Standard Deviation; COPD- Chronic Obstructive Pulmonary Disease; HFrEF Heart failure reduced ejection fraction (≤40%); HFmrEF Heart failure mid-range ejection fraction (41–49%); HFpEF Heart failure preserved ejection fraction (≥50%); NYHA New York Heart Association heart failure classification; ACE-Angiotensin converting Enzymes; ARB-Angiotensin Receptor Blockers; CCB-Calcium Channel Blockers*

**Table 3. A summary of differences in clinical variables between the supervised ET and usual care groups at baseline.**

| | Usual care (mean ± SD) | Exercise therapy (mean ± SD) | Unadjusted F-statistic (P-value) | Adjusted f-statistic (P-value) | Total (mean ± SD) |
|---|---|---|---|---|---|
| BMI (kg.m$^{-2}$) | 28.1±5.2 | 29.3±7.1 | 0.59 (0.45) | 0.97 (0.33) | 28.7±6.2 |
| SBP (mmHg) | 127.1±19.8 | 129.5±16.2 | 0.25 (0.62) | 0.37 (0.55) | 128.3±17.9 |
| DBP (mmHg) | 83.7±12.9 | 82.4±13.8 | 0.14 (0.71) | 0.07 (0.79) | 83.1±13.2 |
| HR (beats.min$^{-1}$) | 78.3±12.6 | 82.8±10.3 | 2.23 (0.14) | 2.47 (0.12) | 80.6±11.6 |
| O$_2$ Sats (%) | 98.4±0.5 | 98.0±0.6 | 9.99 (0.004*) | 9.69 (0.003*) | 98.2±0.6 |
| 6MWD (meters) | 258.8±106.5 | 276.7±64.5 | 0.62 (0.43) | 0.16 (0.70) | 267.8±87.8 |
| VO$_2$peak (ml·kg$^{-1}$·min$^{-1}$) | 11.8±3.2 | 12.3±1.9 | 0.62 (0.43) | 0.16 (0.70) | 12.0±2.6 |

Δ Change; SD standard deviation; F f-statistic from univariate Anova; P-value significance level set at 0.05;

* significant

Forty five percent of all participants were living with HIV-infection. The socio-demographic characteristics of participants randomised into the usual and exercise treatment arms of the study did not significantly differ between the groups as described in Table 2.

## Differences in variables between participants in control and ET treatment groups at baseline

At baseline, participants had an overweight BMI (28.7 ± 6.2 kg.m$^{-2}$), elevated SBP (128.3 ± 17.9 mmHg) and DBP (83.1 ± 13.2 mmHg) and lower functional capacity (6MWD 267.8 ±87.8 meters) and predicted Vo$_2$peak (7.4 ± 0.9 ml·kg$^{-1}$·min$^{-1}$). There were no significant differences in BMI, SBP, DBP, functional capacity and predicted VO$_2$ peak between participants in the usual care and ET arms even after adjusting the results for age, gender, and HIV status (all p>.05). Table 3 describes the summary of the baseline findings.

## Effects of exercise therapy on cardiopulmonary parameters at 12-weeks

Compared to the usual care, participants in the supervised exercise treatment group experienced large, significant reductions in BMI (-0.7 ± 1.1 versus 0.1± 0.6 kg.m$^{-2}$, p = 0.001, $\eta_p^2$ = 0.17) and resting HR (-8.1 ± 12.3 versus 2.8 ± 12.3 beats.min$^{-1}$, p = 0.002, $\eta_p^2$ = 0.16) at the end of 12-weeks ET period. Significant 28.4 and 20.3% increases in measures of functional capacity (103.6 ± 48.2 versus 13.9 ± 41.9 meters, p = 0.001, $\eta_p^2$ = 0.51) and predicted VO$_2$ peak (1.21 ± 0.62 versus 0.14 ± 0.44 ml·kg$^{-1}$·min$^{-1}$, P = 0.001, $\eta_p^2$ = 0.50) were also evident with the ET compared to usual care, respectively. Participants in the exercise intervention group also experienced significant reductions in sub-maximal exercise HR (-5.8 ± 1.13, -5.6 ± 0.74, -3.6 ± 0.48; all P = 0.001) and RPE (-1.6±1.0, -1.2±0.9, -1.4±1.0; all p<0.001) responses to exertion at stages 1, 2 and 3 of Chester step test, respectively.

Those in ET group also experienced significant medium reductions in SBP (-8.7 ± 13.2 versus -0.13 ± 13.4 mmHg, p = 0.02, $\eta_p^2$ = 0.1), DBP (- 6.2 ± 12.4 versus -0.2±10.9, p = 0.05, $\eta_p^2$ = 0.07), and significant moderate increase in O$_2$Sats (0.4 ± 0.7 versus 0.1±0.6%, p = 0.05, $\eta_p^2$ = 0.07) compared to participants in the usual-care. The differences remained significant after adjusting for age, gender, and HIV status (all P <0.05), except for O$_2$Sats (P > 0.05). Table 4 describes the delta/mean differences in the variables between the participants in ET and usual care groups during the supervised, hospital 12 week exercise intervention phase. Figs 2 and 3 illustrate comparison of the pre- and post ET measurements of HR and RPE at each stage of exercise tolerance test among participants in the ET group respectively.

**Table 4. Comparison of the 12-weeks changes in clinical variables between the ET and usual care study groups.**

| | Usual care Mean Δ±SD (%Δ) | Exercise therapy Mean Δ±SD (%Δ) | Unadjusted p-value($\eta_p^2$) | Adjusted p-value($\eta_p^2$) |
|---|---|---|---|---|
| BMI (kg.m$^{-2}$) | 0.08±0.6 (0.36) | -0.69±1.06 (2.2) | 0.001*(0.17) | 0.001*(0.17) |
| SBP (mmHg) | -0.13±13.4 (0.42) | -8.69±13.17 (5.6) | 0.02*(0.10) | 0.03*(0.09) |
| DBP (mmHg) | -0.17±10.9 (0.98) | -6.24±12.41 (4.0) | 0.05*(0.07) | 0.05*(0.07) |
| Resting HR (beats.min$^{-1}$) | 2.80±12.9 (5.63) | -8.10±12.34 (8.5) | 0.002*(0.16) | 0.004*(0.15) |
| O$_2$ Sats (%) | 0.07±0.6 (0.07) | 0.41±0.68 (0.4) | 0.05*(0.07) | 0.06(0.07) |
| 6MWD (meters) | 13.93±41.9 (9.21) | 103.59±48.19 (38.6) | 0.001*(0.51) | 0.001*(0.51) |
| VO2peak (ml·kg$^{-1}$·min$^{-1}$ | 0.42±1.3 (5.0) | 3.04±1.46 (25.3) | 0.001*(0.49) | 0.001*(0.49) |

Δ Change; SD standard deviation; p-value significance level set at 0.05; $\eta_p^2$ effect size;

* Statistically significant P-value

## Effects of exercise therapy on cardiopulmonary parameters across the study follow-up period

The summary statistics of the mixed model analyses including type III tests of fixed effects for groups (ET versus usual care), time (baseline to final follow-up at 24 weeks) and interaction

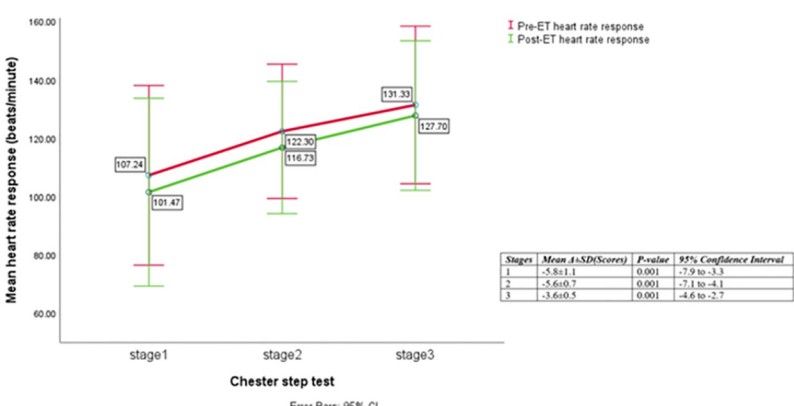

**Fig 2. Comparison of mean (SD) heart rate responses at the end of the three standardized 2-minute stages of the Chester step test before and after ET within the intervention study group (n = 30).**

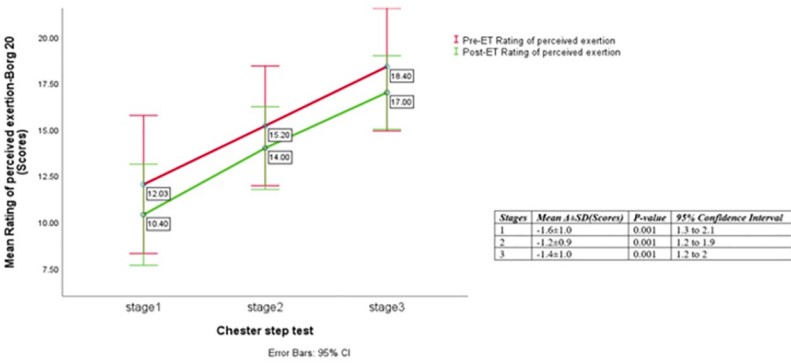

**Fig 3. Comparison of mean (SD) rating of perceived exertion score responses at the end of the three standardized 2-minute stages of the Chester step test before and after ET within the intervention study group (n = 30).**

**Table 5. Summary statistics, including Type III Tests of Fixed Effects, from the mixed model analyses for the study main and secondary outcome variables.**

| Dependent Variables | TYPE III Tests of Fixed Effects | | | | | |
|---|---|---|---|---|---|---|
| | 6MWD (m) | VO$_2$peak (ml.kg.min$^{-1}$) | Resting HR (beats.min$^{-1}$) | O$_2$Sats (%) | SBP (mmHg) | DBP (mmHg) |
| GROUPS | p< 0.001* | p< 0.001* | P = 0.453 | P = 0.357 | P = 0.44 | P = 0.046* |
| TIME | p< 0.001* | p< 0.001* | P = 0.121 | p<0.001* | P = 0.17 | P = 0.051 |
| GROUP versus TIME | p< 0.001* | p< 0.001* | P = 0.022* | P = 0.016* | P = 0.12 | P = 0.161 |
| Assessment observations completed | IT n = 144<br>UC n = 144 | IT n = 144<br>UC n = 144 | IT n = 142<br>UC n = 143 | IT n = 142<br>UC n = 144 | IT n = 142<br>UC n = 144 | IT n = 142<br>UC n = 144 |
| Akaike's Information Criterion (AIC) | 2952.3 | 474.9 | 2085.5 | 443.6 | 2231.8 | 2140.7 |

effects for the main study variables over the study are reported in Table 5 below. Significant group versus time interaction effects were observed in 6MWD (p<0.001), predicted Vo$_2$peak (p<0.001), O$_2$Sats (p = .016) and resting HR (p = 0.022) between participants in the usual care and ET arms of study over the five time points (from baseline through to the two months post ET period). For the primary study outcomes, 6MWD and predicted VO$_2$peak increased progressively over time compared to the usual care group and pairwise comparisons showed that they were significantly different from the 8- week time point to the final assessment at 24 weeks (all P-values for interaction effect, p< 0.001). With the ET group, based on the post hoc comparisons of the estimated marginal means, only the 16-to-24-week period failed to show a significantly higher 6MWT distance (12.6m). Consistent with the fitness improvements, a reduction pattern was observed in resting HR over time among the participants in the ET group compared to non-significant changes in the usual care group. The difference in resting HR between the groups was statistically significant at the end of the supervised ET phase (pairwise comparison p = 0.029) but was not retained thereafter. Likewise, a significant increase was observed in O$_2$Sats with ET—with no change evident in the usual care group. The significant changes in O$_2$Sats were noted during the early supervised ET period (8week compared to baseline). There were significantly lower DBP values in the group over the study period-with a tendency for both groups to lower DBP values over the study period. However, no significant effects of time or interaction effects with exercise therapy were observed for either SBP and DBP changes across the follow-up period. Table 6 shows the summary statistics from the

**Table 6. Comparison of change differences between the ET and usual care groups across the follow-up timepoints.**

| Variables | MEAN ± SD (95% CI) | | | | |
|---|---|---|---|---|---|
| | Baseline | 8-weeks of supervised ET | End of supervised ET (week 12) | One-month post supervised, continuing home-based ET (week 16) | Two-months post supervised, continuing home-based ET (week 24) |
| Resting HR (beats.min$^{-1}$) | | | | | |
| Control | 78.3±11.5 (74.1–82.5) | 80.0±13.2 (75.1–84.9) | 81.1±10.4 (77.2–85.0) | 76.6±13.2 (71.5–81.7) | 80.3±12.2 (75.4–85.2) |
| Treatment | 82.8±11.5 (78.6–87.0) | 76.4±13.2 (71.5–81.3) | 74.9±10.96 (71.0–78.9) | 75.6±13.8 (70.4–80.7) | 76.9±12.2 (72.1–81.7 |
| O$_2$Sats (%) | | | | | |
| Control | 98.4±0.6 (98.2–98.6) | 98.6±0.5 (98.4–98.8) | 98.5±0.5 (98.3–98.7) | 98.6±0.5 (98.5–98.8) | 98.7±0.4 (98.5–98.8) |
| Treatment | 98.0±0.6 (97.8–98.2) | 98.5±0.5 (98.3–98.7) | 98.4±05 (98.2–98.6) | 98.7±0.5 (98.5–98.9) | 98.8±0.4 (98.7–99.0) |
| SBP (mmHg) | | | | | |
| Control | 127±19.2 (120.5–133.7) | 126.0±21.4 (118.0–133.4) | 127.0±17.0 (120.7–133.3) | 129.0±15.9 (123.1–135.0) | 130.0±14.3 (124.0–135.4) |
| Treatment | 129.5±19.2 (122.9–136.1) | 123.8±21.4 (116.1–131.5) | 121.8±17.5 (115.4–128.1) | 121.4±15.9 (115.4–127.4) | 127.0±14.3 (121.4–132.7) |
| DBP (mmHg) | | | | | |
| Control | 83.7±13.2 (78.9–88.6) | 81.4±13.2 (76.6–86.2) | 83.6±10.4 (79.8–87.3) | 82.4±10.6 (78.4–86.4) | 87.1±14.3 (81.5–92.7) |
| Treatment | 82.4±13.2 (77.6–87.3) | 79.8±13.2 (74.9–84.6) | 76.0±10.4 (72.2–79.8) | 76.2±10.6 (72.1–80.3) | 79.9±14.3 (74.3–85.6) |

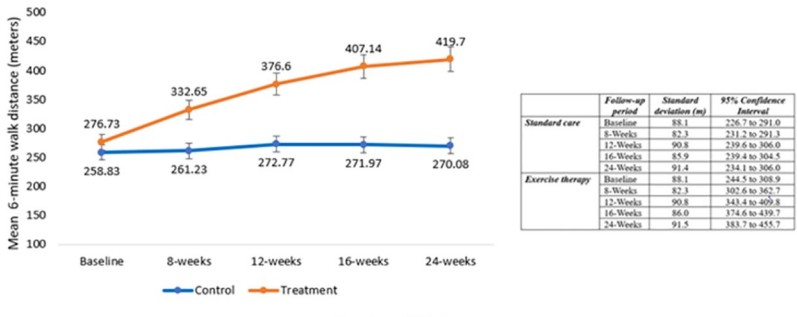

**Fig 4. Comparison of changes in 6MWD between participants in the usual care and ET arms across the five follow-up timepoints.** 6MWT measures at all time points 8 to 24 weeks were significantly different versus usual care (p-value for interaction effects p = or < 0.001).

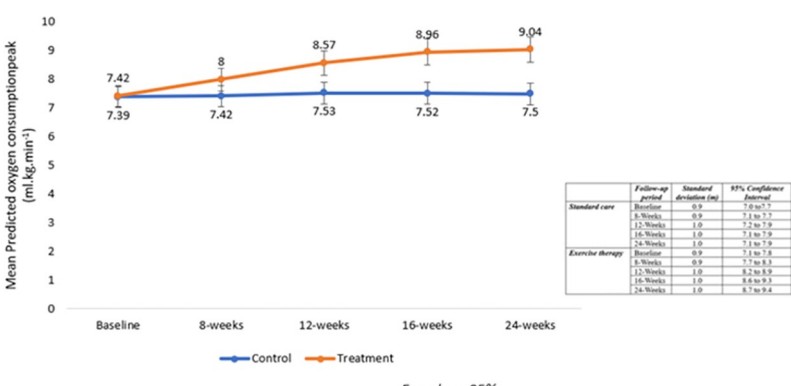

**Fig 5. Comparison of change differences in predicted VO$_2$peak between participants in the usual care and ET arms across the five follow-up timepoints.** Mean peak VO$_2$ measures at all time points 8 to 24 weeks were significantly different versus usual care (p-value for interaction effects p = 0.012 (baseline versus week8) and P < 0.001 at all later time points).

mixed model analyses for the study main and secondary outcome variables 6MWT, VO2peak, with resting HR, O$_2$Sats, SBP, and DBP. Figs 4 and 5 illustrate the changes in the 6MWD and predicted Vo$_2$peak over time respectively between the ET and usual care groups, including the supervised and home-based periods.

## Participants' reported barriers to performing physical activity at home

Only 46.7% of the participants in the ET arm attended > 90% of the total hospital-based exercise sessions. Eight weeks into the intervention, participants in the ET and usual care study arms reported musculoskeletal and(or) chest pains (n = 5, 4), lack of motivation (n = 0, 8), lack of family support (n = 12, 15), fatigue-related unproductivity post exertion (n = 6, 0), fear (n = 0, 7) and insomnia post exertion (n = 1, 0) as factors that hindered them from engaging in PA at home respectively. At 12-weeks, lack of motivation (n = 0, 4) and family support (n = 5, 20), and fatigue-related unproductivity (n = 2, 0) were also reported as barriers to engaging in PA, respectively.

### Frequency of hospitalization during the study and adverse effects of the intervention

No participant in either of the study arms was hospitalized from the enrolment to the end of three months follow up period post intervention. One death occurred among participants in the usual care arm within the 16-weeks of follow-up, and the death was due to septicaemia. No adverse effects related to exertion were recorded in this study.

### Healthcare workers' reported challenges to delivering CR at QECH

Evaluation of the Physiotherapists who facilitated CR delivery reported the following main themes (i) inadequate space and under resourced multipurpose rehabilitation gymnasium, (ii) uncoordinated patient referral system, and (iii) inadequate human resource as factors that limited delivery of CR intervention at the hospital.

## Discussion

The CHF participants recruited to this study were a heterogeneous cohort, largely overweight, with a history of hypertension, and as might be expected, showed markedly low functional capacity and CRF outcomes at baseline. The participants exhibited some typical clinical features of stable, non-hospitalised HFpEF (including severe exercise intolerance, normal or mid-range ejection fraction, > 90% on maintenance diuretics). The randomised ET intervention was effective in reducing blood pressure, BMI, and HR, and increasing measures of exercise tolerance compared with usual care. ET participants also exhibited reducing exercise-induced heart rates and perceived exertion with improved CRF. The continued improvement in functional capacity and CRF outcomes were also observed in the ET group within the maintenance phase. Whereas no meaningful improvements within groups were observed in blood pressure, BMI, $O_2$ Sats and HR during this time.

### Baseline characteristics of participants

The age, gender, BMI, and cardiac function (EF) did not differ significantly between participants in usual care and ET groups. The mean age in our study is comparable with other studies undertaken in SSA, including Malawi [29, 30] and Nigeria [31–33], and corresponds with the suggestion of CHF occurring between 50 and 60 years of age among native African populations [34]. Majority of participants in our recruited CHF cohort were women (56.7%), which is consistent with other investigators [29, 30, 33]. HFpEF accounted for 70% of CHF cases in the study of Ajiboye et al. [33] within a tertiary hospital in Nigeria. Hypertension was the underlying CVD in over 80% of the subjects within this group and females accounted for over 60% of the consecutively recruited subjects [33]. Similar to Ajiboye et al. [33], participants in this current study were overweight, over half (56%) had HFpEF and were classified in NYHA class III (65%). It has been suggested that hypertension may be an important underlying factor for heart failure symptoms in native African populations [33]. Contrary, recent findings from a study based in Neno, rural Malawi, showed more heterogeneous aetiology of CHF symptoms (with the most common diagnostic categories by cardiac ultrasound), namely hypertensive heart disease (36%), cardiomyopathy (26%), and rheumatic, valvular or congenital heart disease (12.3%) [29]. Likewise, CHF patients in Malawi are more likely to be older, have normal BMI (18–24 kg.m$^{-2}$) [29], be classified in NYHA II (58–64%) [29, 30] and have reduced LVEF (55%) [30].

The differences in findings on BMI and NYHA classes between this current study and Mailosi et al.'s [29] could possibly be attributed to the fact that most of the participants in our

study were from the urban settings of Blantyre and were referred to this tertiary hospital for specialist care due to severity of their symptoms, respectively. Higher proportions of participants in our study also had prior diagnosis of hypertension (83% versus 44%), Type 2 diabetes (18% versus 2.8%) and were living with HIV (45% versus 10.7%) compared to participants in Mailosi *et al.* [29]; which further explains the variation and the implications of urban versus rural settlement on non-communicable diseases [10, 35]. Nevertheless, Mwabutwa et al. [30] also conducted their study at a tertiary hospital. Variation in the proportion of participants living with HIV-infection in the current study (45%) and Mwabutwa et al.'s [30] (15%) may explain the difference in the presentation of cardiac dysfunction (reduced versus preserved LV EF) [36–38].

The findings of the current study also indicate that the participants had remarkably low predicted $VO_2$ peak (12 ml.kg$^{-1}$.min$^{-1}$) at baseline. Correspondingly, investigators who have assessed CRF in patients with HFpEF using gold standard cardiopulmonary exercise testing procedures have also reported similarly low values of $VO_2$ peak of between 13–14 ml.kg$^{-1}$.min$^{-1}$ [18, 39–41]. Similarly, the short walked distance (6MWD) recorded in the current study (mean 268 m) is somewhat lower than reported within earlier findings (273-318m) [42, 43] and may correspond with referral for severity of symptoms. Participants recruited to Wolsk *et al.* [43] and Forsyth *et al.*[42] investigations were considerably older (67–79 years).

## Effects of ET on blood pressure and heart rate

In the present study, ET caused significant reductions in resting SBP and DBP (-8.7, -6.0 mmHg, respectively) compared with the usual lifestyle change targeted education and counselling during the supervised phase of the ET intervention period. Similarly, a 6.0 mmHg reduction in SBP was reported among heart failure patients who received ET for 12-weeks in Nigeria [33]. Our findings are also supported by Whelton *et al.*'s [44] observation that reductions in SBP with ET are more pronounced among people of African origin compared to other ethnicities.

Contrary, previous study in Nigeria reported non-significant reductions in resting SBP and DBP with ET [45]. This could be due to the difference in the ET dosage; although participants in their study received hospital-based ET three times a week, unlike in our study (once and twice a week at hospital and home respectively). In the above study, ET mainly constituted static-cycling whereas participants in our study also performed various total-body movement exercise types, such as stepping, dancing, ball-throwing, treadmill walking and weight-lifts, plus stretches which were delivered using interval circuit approach. It is imperative to note that positive change in lifestyle behaviour, and associated weight loss, may as well attribute to the observed improvement. Indeed, our findings show that ET promoted significant reduction in BMI compared with usual care. However, the reduction did not change the overweight status of the patients at 12-weeks. Similar reductions were also reported among cardiac patients in Egypt [46].

Participants in the ET group also experienced a substantive (- 8.0 beats.min$^{-1}$) reduction in resting HR compared to those in the usual care group (3.0 beats.min$^{-1}$ increase), and the reduction pattern was also observed across all five follow-up time points. These findings correspond with what other investigators have reported [33, 45]. Therefore, ET prescription should be done in consideration with CHF medications, especially beta blockers and resting HR adaptations to ET, to ensure the safety of the patients. Appropriate supervision of home-based ET should also be emphasized. Notably, no significant changes in SBP and DBP were observed across the five follow-up timepoints. This implies that effective strategies are required to sustain the cardiovascular-related ET benefits. For instance, long-term PA habits appear

paramount and should be promoted among cardiac patients to sustain the improvement in resting hemodynamic parameters.

### Effects of ET on functional capacity and cardio-respiratory fitness

Although the changes in the 6MWD (104 m) and predicted $VO_2$ peak (3.0 ml·kg$^{-1}$·min$^{-1}$) were significantly higher in the ET compared to the usual care groups, the latter also experienced clinically meaningful change in 6MWD (14m) in this study. Combining both ET and lifestyle targeted education and counselling in CR as it is widely promoted [12, 21], appear a very effective strategy for treating CHF. A 10m increase in walking distance has been associated with nearly 2% reduction in three- and half-year risk for CVD-related mortality [47]. In their findings from a multicentre study, in which 14% of participants were blacks, Matsumoto *et al.* [47] identified 200 m as an optimal cut-off point for predicting cardiac events; which corresponds with findings from other worldwide investigators [48–50]. Therefore, our findings suggest that ET, delivered in this resource-limited setting, may significantly reduce the risk for future cardiac events within higher risk groups and patients with CHF. In the study of Nilsson et al. [15], which assessed the long-term effects of a group-based high-intensity CR model within older CHF patients with reduced EF (30%), categorized in NYHA classes II and III, an average increase of 58 m was reported. This increase in walking distance was maintained as a trend at one-year follow up [15]. The findings supported the recommendation a group-based aerobic interval training programme to improve long-term effects on functional capacity and QoL in patients with CHF.

In contrast, recent RCT findings from the 'OptimEx-Clin' study did not support either high-intensity interval training, or moderate continuous training, compared with guideline-based PA for CHF patients with preserved cardiac function [51]. This multicentre trial with three groups examined different exercise intensities in patients with HFpEF over three-months at the hospital- followed by nine-months of tele-medically supervised home-based exercise training. Interestingly, neither exercise intervention groups met their priori–defined minimal clinically important difference in $VO_2$peak (2.5 ml·kg$^{-1}$·min$^{-1}$) compared with the usual care group at any time point [52]. Although this suggests that neither of the two ET programmes are effective in improving CRF in patients with HFpEF, half of the participants attended only 70% of the ET sessions. Our analyses indicate better compliance with once weekly supervised group-based exercise sessions and improvements in exercise capacity and predicted $VO_2$peak.

In their analysis of results from 'HF-ACTION' RCT with a large sample size within highly developed healthcare settings (n >1000 CHF patients), Swank et al. [23] assessed the relationship between the change in $VO_2$peak on mortality and hospitalization. The group-based progressive aerobic ET delivered three times a week for three months within a hospital setting followed by five-times a week for a long-term period at home, and the standard care constituted lifestyle education booklet with information on health behaviour change [53]. Swank et al. [23] reported that nearly 1ml.kg$^{-1}$.min$^{-1}$ increase in $VO_2$peak led to 8 and 5% reductions in CVD-related deaths or HF-related hospital admissions and time to all-cause deaths or hospitalizations respectively [23]. Therefore, it is reasonable to postulate that the ET in this current study was extremely efficacious in reducing the risk for cardiac and non-cardiac events in this symptomatic group of patients, improvement which was observed even in the maintenance phase. In terms of improvements in cardiorespiratory and perceptual responses to stepping exertion; the changes in HR (-5.8, -5.6, -3.6 beats/minute) and RPE (-1.6, -1.2, -1.4) scores at stages 1, 2 and 3, respectively, of standardised fitness testing in the current study further supported the efficacy of ET in improving cardiorespiratory capacity. Similar reductions in HR (-3.0, -5.0, -4.0 beats/minute) were also reported by Grove, Jones and Connolly [54] who

retrospectively assessed fitness of 169 older adults (aged 66.8 ± 7.3 years) with high risk cardiac or heterogenous CVD states. In that study, the Chester step test was conducted before and after the 12-weeks supervised aerobic circuit interval with low-intensity strength training, which was offered three times a week. Following the intervention, $VO_2$ peak also increased by 2.8 ml.kg.$^{-1}$.min$^{-1}$ and an overall mean 4.1 beats.min$^{-1}$ (p≤0.001) reduction in the exercise HR response on the CST was observed [54].

In CHF, impaired functional capacity and poor CRF are attributed to several integrated physiological factors such as cardiac dysfunction, ventilatory inefficiency, muscle atrophy and endothelial dysfunction [55]. Normally, the vascular system responds with vasodilation to exercise, to adequately facilitate energy production to meet the exertion demands. The opposite happens with CHF; vasoconstriction impedes circulation to the skeletal muscles including the myocardium, which also impairs the neurovascular system responses [56]. Consequently, CHF patients experience early fatigue, shortness of breath and reduced exercise tolerance, which limits their functional capacity. The observed ET-associated improvements in the walking distance and $VO_2$ peak could be attributed to the increase in peripheral mitochondrial function [56] and lower neurohormonal stimulation [57, 58] and ventilatory efficiency [59], including improvement in endothelial dysfunction and vascular-flow resistance. Strength training delivered as a solo or in combination with aerobic exercises was previously shown to result in meaningful increases in peripheral muscular power [60], $VO_2$ peak and 6MWD [60, 61], which further explains the improvement in mitochondrial function and cardiovascular plus respiratory efficiency.

Although clinically meaningful increases in the walking distance was observed across the five follow-up timepoints among participants who received ET compared to the usual care in this study, the consecutive improvements in functional capacity and CRF plateaued in the final phase of the maintenance period as shown on Figs 3 and 4. Hence, there is a requirement to institute long-term strategies to enhance continued effective self-care management. Nonetheless, the overall improvements in functional capacity and estimated cardiorespiratory fitness induced by this hybrid exercise training protocol were remarkable and clinically highly relevant compared to usual care. According to Fernhall [62], long-term ET should be promoted among cardiovascular patients to sustain the benefits. In this resource-constrained setting, caregivers should be intensely supported/educated to facilitate home-based rehabilitation, and physiotherapists should be engaged in the routine monitoring of patients to promote life-long self-care and PA engagement. Promoting multidisciplinary teamwork in intensive supervised delivery and caregivers thereafter in home-based maintenance of exercise training in the management of cardiovascular and CHF patients in this setting is key.

## Barriers to engaging in physical activity at home

Although caregivers for participants in ET and usual care arms received training to facilitate home-based ET and lifestyle modification respectively, some patients missed their scheduled supervised ET sessions and found it difficult to modify their lifestyle behaviours. Lack of time for both patients and caregivers, patients' fear to engage in PA, and lack of motivation, were reported as contributing factors. An expert panel review indicated that in the highly supervised 'HF-ACTION' clinical trial -although exercise equipment was provided at home and intensive efforts were made to promote compliance among CHF patients enrolled, long-term adherence was poor at < 30% [63]. Similar barriers were also reported by CR attendees in Kenya and Nigeria [33, 64]. Although home-based rehabilitation approach were suggested to be effective, adherence has been previously reported to be poor [64]. In low-resource settings of SSA, our study has demonstrated the home-based approach is feasible and realistic. However,

appropriate, and enhanced measures need to be instituted to maximize adherence and safety and linked well with supervised sessions. For instance, some participants in the current study also reported musculoskeletal or chest pains as limiting factor to ongoing PA. In relation to this, it should be noted that the number of adverse and serious adverse events were reported as high within the 'OptimEx-Clin' study and was interpreted to reflect the multimorbid condition of the CHF patients recruited [52]. The higher number of non-serious, non-cardiovascular adverse events in the training groups may be explained by the more frequent contacts and therefore higher reporting in these groups (e.g., number of respiratory tract infections and knee/hip pain). Routine monitoring of ET dosage and frequent education on exertion intensity and self-guidance also helped in resolving such factors and enhanced safety in the current study. As outlined by Azevedo and Santos [65], ET prescription should consider the underlying cause of CHF, medications used, left ventricular EF, NYHA classification, musculoskeletal disorders and ischaemia grades or presence of arrythmia in ensuring exercise safety and efficacy of PA in CHF patients.

## Barriers to delivering CR

Inadequate infrastructure and human resource emerged as key institutional barriers to delivering CR in this study. At QECH, rehabilitation services are delivered to both in- and out-patients within the same facility (Physiotherapy department) which is relatively small, has one gymnasium and is reported by professional staff as "under-resourced". Similar challenges were also reported within structured exercise training studies in Kenya [64]. Promoting multidisciplinary teamwork in CR may assist in finding long-term solutions that may promote uninterrupted and effective CR delivery in this setting. Need for capacity building among team members is also warranted in this setting.

## Strengths and limitations

This is the first study to assess effects of ET in patients with CHF in Malawi and has used a novel CR delivery approach within this environment which may feasibly be adapted in this resource-constrained nation. Nevertheless, the findings should be interpreted with caution. We used a submaximal field test (6MWT) and a predictive equation to assess functional capacity and CRF, respectively. The walking speed is self-paced by the participant and familiarity may affect the performance [66] if the procedure is not understood. Nevertheless, researchers thoroughly explained and demonstrated the procedure to each participant. Also, because the participants already had symptoms, such as dyspnoea and shortness of breath with ordinary activities, the chances of underestimating the functional capacity were not considered to be substantial. Literature suggest that the test is well correlated (moderate-good) with the oxygen consumption measured using ergometer-cycling in CHF [67, 68]. Although Cahalin et al. [25] equation has the best $VO_2$peak predictive validity compared to other existing applicable equations [26], the race of the participants used in developing such equation was not described. It is known that healthy African black individuals naturally have several physiological characteristics such as lower lung volume [69, 70], high bone mineral content and density and limb-mass and muscle fibre characteristics [71] which may translate to lower estimated maximal oxygen consumption from stepping exercise compared to other races. A small sample size used in this study also limits generalizability of the findings, hence the need to conduct large sample sized studies in this setting. Future studies should also consider comparing the effects of ET delivered fully at the hospital and at home or in the community.

## Conclusion

This novel study has shown that a short-term hybrid model of supervised ET- with an integrated delivery of once and twice a week within hospital and home settings, respectively, significantly improves measures of cardiorespiratory function, including functional capacity of patients with CHF. Improved CRF was demonstrated by lower cardiovascular and perceptual responses to exertion using graded submaximal exercise testing, feasible for the low resource environment. Structured exercise intervention was also shown to significantly improve measures of BMI and resting haemodynamic status. Accordingly, ET can feasibly be delivered to CHF patients within integrated low-resource hospital and home-based carer supervised settings within SSA countries, such as Malawi. Appropriate measures should be instituted to enhance adherence to home-based structured PA and safety including education peer supervision. Also, to maximize access to and effectiveness of CR intervention, multidisciplinary teamwork within CR designing and implementation and both aerobic and resisted exercise components should be promoted. If home-based CR approach is used to promote PA engagement, patients with CVD should routinely be periodically monitored through local tertiary facilities to evaluate programme efficacy and safety of approach to prescribing exercise training intensity. Future research should compare effects of hospital- and home-based CR delivery approaches within Malawi, and wider sub-Saharan African settings and investigate feasible strategies to enhance continuity of self-care post discharge from CR.

## Supporting information

**S1 Checklist.**
(PDF)

**S1 Dataset.**
(XLS)

**S1 File.**
(PDF)

## Acknowledgments

We would like to acknowledge contributory works of Mr. Emmanuel Maluza, Mr. Felix Kaphwiyo and Mr. Vincent Samuel Phiri from the Kamuzu University of Health Sciences for the logistical, technical, and statistical support rendered towards the implementation of this study, respectively.

## Author Contributions

**Conceptualization:** Alice Namanja, Daston Nyondo, Tendai Banda, Ephraim Mndinda, Johnstone Kumwenda.

**Data curation:** Alice Namanja.

**Formal analysis:** Alice Namanja, Adrian Midgely, James Hobkirk, Sean Carroll, Johnstone Kumwenda.

**Funding acquisition:** Alice Namanja.

**Investigation:** Alice Namanja, Daston Nyondo, Tendai Banda, Ephraim Mndinda, Adrian Midgely, Johnstone Kumwenda.

**Methodology:** Alice Namanja, Adrian Midgely, Johnstone Kumwenda.

**Project administration:** Alice Namanja, Daston Nyondo, Johnstone Kumwenda.

**Resources:** Alice Namanja, Daston Nyondo, Tendai Banda, Ephraim Mndinda, James Hobkirk, Sean Carroll, Johnstone Kumwenda.

**Software:** Adrian Midgely, Sean Carroll.

**Supervision:** Alice Namanja, Johnstone Kumwenda.

**Validation:** Alice Namanja.

**Visualization:** Alice Namanja.

**Writing – original draft:** Alice Namanja, James Hobkirk, Sean Carroll, Johnstone Kumwenda.

**Writing – review & editing:** Alice Namanja, Daston Nyondo, Tendai Banda, Ephraim Mndinda, James Hobkirk, Sean Carroll, Johnstone Kumwenda.

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
