## [Decision Letter · Decision Letter 0]

5 Oct 2023

PONE-D-23-25863Delivering effective, comprehensive, multi-exercise component Cardiac Rehabilitation (CR) for chronic heart failure patients in low resource settings in sub-Saharan Africa: Queen Elizabeth Central Hospital - (QECH-CR) randomised CR study, Malawi.PLOS ONE

Dear Dr. Namanja,

Thank you for submitting your manuscript to PLOS ONE. After careful consideration, we feel that it has merit but does not fully meet PLOS ONE’s publication criteria as it currently stands. Therefore, we invite you to submit a revised version of the manuscript that addresses the points raised during the review process.

We look forward to receiving your revised manuscript.

Kind regards,

Yoshihiro Fukumoto

Academic Editor

PLOS ONE

Journal Requirements:

   "The first author (Alice Namanja) received a small research grant from the Malawi NCD BRITE consortium to implement this project. The funders also offered research writing skill development and implementation science. NCD BRITE Consortium assigned Alice Namanja to a local expert in the field of Cardiovascular disease prevention and management. Together, with other identified collaborators, they developed the project, implemented it and prepared the manuscript."

Reviewers' comments:

Reviewer's Responses to Questions

**Comments to the Author**

1. Is the manuscript technically sound, and do the data support the conclusions?

Reviewer #1: Partly

Reviewer #2: Yes

2. Has the statistical analysis been performed appropriately and rigorously? 

Reviewer #1: No

Reviewer #2: Yes

3. Have the authors made all data underlying the findings in their manuscript fully available?

Reviewer #1: No

Reviewer #2: Yes

4. Is the manuscript presented in an intelligible fashion and written in standard English?

Reviewer #1: Yes

Reviewer #2: Yes

5. Review Comments to the Author

Reviewer #1: PONE-D-23-25863: statistical review

SUMMARY

This is a randomized study to estimate the effect of cardiac rehabilitation (CR) on a battery of biometrical outcomes. After a baseline screening, subjects were randomly allocated to a CR treatment group and to a standard care group. Subjects in the treatment group were treated up to 12 weeks. Both groups were screened after 8, 12, 16 and 24 weeks. There are not relevant groups effects at baseline (Tables 2 and 3), indicating a good randomization. For each biomarker of interest, treatment effects were tested on the difference between the outcome values at two different points in time. All the statistical analysis relies on ANOVA models that could be replaced by more standard methods (see major issue 1). I’m also listing below some specific points that need some clarifications.

MAJOR ISSUES

1. Although repeating multiple ANOVAS for comparing outomes at pairs of points in time is technically correct in a longitudinal study, differences at two points in time can be computed only when both values are available. Unfortunately, longitudinal studies are often affected by drop-outs and this study is not an exception (as shown by Table 1). Asa result, ANOVAs are performed on complete trajectories, by discarding partial trajectories. This waste of information is unnecessary, because ANOVA tables can be replaced by a single mixed regression model that includes a random effect at the subject level and is estimated by including full and partial individual trajectories. Mixed models are the state-of-the-art method for longitudinal data analysis and are available in SPSS, which is the software used by the authors.

SPECIFIC POINTS

1. Table 2. It is good that p-values are large, as they support the randomness of treatment allocation. However, why are the p-value preceded by the symbol “<”?

2. It looks like the values observed at the 8th week have not been included in the analysis. Why? Aren’t the short-term effects of the treatment worth investigation?

3. I did not understand whether Table 5 is about differences with respect the 12th week or the baseline. Please clarify.

Reviewer #2: Dear Authors,

Thank you very much for the opportunity to review your work on “Delivering effective, comprehensive, multi-exercise component cardiac rehabilitation (CR) for chronic heart failure patients in low resource settings in sub-Saharan Africa: Queen Elizabeth Central Hospital – RCT study, Malawi.” This is an RCT study of cardiac rehabilitation implemented on a novel hybrid exercise training model vs conventional usual care. A total of n=60 were recruited from baseline, and eventually, n=50 completed all data points at the end of 24 weeks.

I wish to congratulate the authors for their contributions to the field. However, I have some feedback/concerns before the manuscript should be considered ready for publication.

Introduction:

Page 4 line 90-95: Please standardise reference styles used. Also, a few unidentified symbols should be corrected before submission.

Page 5 line 102: Please clarify the abbreviations for QE CR, which I presume stand for Queen Elizabeth and cardiac rehabilitation; it is probably a good idea to provide the full terms before abbreviations are used.

Methods:

Page 5, lines 108, 109, and 114. Similar to those mentioned above, please provide the full term for QECH, UMIN-CTR, and BACPR.

Page 7, line 147: (r)elated data.

Page 7, line 169: I suggest adding a quick line of explanation as to why a 20-m long corridor was used for 6MWT, while ATS (2003) recommended a minimum of 25/30 metres. Limitations happen in clinical settings; maybe a short note to close the loop for the benefit of future readers.

Furthermore, please elaborate if participants were required to undergo the 6MWT once or twice. The information is missing.

Page 8, lines 184-186: Incomplete statement

Page 8, lines 186-187: please further elaborate on how the 1RM testing was performed.

Page 9, line 195: maybe replace the word guardians with caregivers? Given that the participants are grown-up adults.

Methods in general: please elaborate on how the data/result corresponding to “healthcare workers’ reported challenges to delivering CR at QECH” were collected. The whole section is missing from the Methods.

Results:

What is the final statistical power with n=60? That differed from the original sample size calculation of n=80.

Figure 2 & 3: maybe suggest revising the legends of the graph from “pre-heart rate response” and “post-heart rate response” to “Pre-ET heart rate response” and “Post-ET heart rate response.” I found it confusing initially, uncertain if it refers to pre-post chest step test or pre-post exercise training (CR).

Discussion: it would be interesting if the authors could discuss the intermediate long term training effects of this novel hybrid CR, which is currently missing. I read that the improvement of 6MWD in the ET group improved by 103.59+/-48.19 metres (table 4). However, the comparison of the change in 6MWD during the maintenance phase 30.36+/-31.3 and 43.53+/-73.8 metres at 1- and 3-months follow-up respectively.

6. PLOS authors have the option to publish the peer review history of their article (what does this mean?). If published, this will include your full peer review and any attached files.

Reviewer #1: No

Reviewer #2: No

---

## [Author Response · Author response to Decision Letter 0]

22 Dec 2023

Dear Editor, 

Thank you very much for the feedback we received from the reviewers. We would like to resubmit our revised manuscript based on the feedback we received. We would like to also mention that the funder (Malawi NCD BRITE consortium) had no role in study design, data collection and analysis, decision to publish, or preparation of the manuscript.

Below are our responses to the reviewers’ comments. 

REVIEWER 1:

SUMMARY

This is a randomized study to estimate the effect of cardiac rehabilitation (CR) on a battery of biometrical outcomes. After a baseline screening, subjects were randomly allocated to a CR treatment group and to a standard care group. Subjects in the treatment group were treated up to 12 weeks. Both groups were screened after 8, 12, 16 and 24 weeks. There are not relevant groups effects at baseline (Tables 2 and 3), indicating a good randomization. For each biomarker of interest, treatment effects were tested on the difference between the outcome values at two different points in time. All the statistical analysis relies on ANOVA models that could be replaced by more standard methods (see major issue 1). I’m also listing below some specific points that need some clarifications.

MAJOR ISSUES

REVIEWER: repeating multiple ANOVAS for comparing outcomes at pairs of points in time is technically correct in a longitudinal study, differences at two points in time can be computed only when both values are available. Unfortunately, longitudinal studies are often affected by drop-outs and this study is not an exception (as shown by Table 1). Asa result, ANOVAs are performed on complete trajectories, by discarding partial trajectories. This waste of information is unnecessary, because ANOVA tables can be replaced by a single mixed regression model that includes a random effect at the subject level and is estimated by including full and partial individual trajectories. Mixed models are the state-of-the-art method for longitudinal data analysis and are available in SPSS, which is the software used by the authors.

RESPONSE: We have incorporated mixed model regression analysis to analyse the data, especially when looking at the entire follow-up timepoint data. We have maintained Anova analyses to assess the direct effects of ET, by comparing the baseline and end of ET outcomes; there was no drop-out in the trial phase, though the sample size was less than the calculated one due to COVID-19 related participant enrolment challenges.

SPECIFIC POINTS

REVIEWER: Table 2. It is good that p-values are large, as they support the randomness of treatment allocation. However, why are the p-value preceded by the symbol “<”?

RESPONSE: The p-value presentation has been corrected throughout

REVIEWER: It looks like the values observed at the 8th week have not been included in the analysis. Why? Aren’t the short-term effects of the treatment worth investigation?

RESPONSE: There were no significant changes at 8-weeks, hence skipping it. However, we have included the 8-weeks in the mixed model regression analysis.

REVIEWER: I did not understand whether Table 5 is about differences with respect the 12th week or the baseline. Please clarify.

RESPONSE: The table 5 has been removed from the manuscript, replaced by the results from the mixed model analysis.

REVIEWER 2:

REVIEWER: Page 4 line 90-95: Please standardise reference styles used. Also, a few unidentified symbols should be corrected before submission. 

RESPONSE: The references have been modified (see the reference links colored in yellow). The unidentified symbols have also been rectified (i.e HF change to heart failure).

REVIEWER: Page 5 line 102: Please clarify the abbreviations for QE CR, which I presume stand for Queen Elizabeth and cardiac rehabilitation; it is probably a good idea to provide the full terms before abbreviations are used. 

RESPONSE: QECH-CR has been written in full first before the abbreviation (See page 5 line 97- yellow colored sentence)

Methods:

REVIEWER: Page 5, lines 108, 109, and 114. Similar to those mentioned above, please provide the full term for QECH, UMIN-CTR, and BACPR. 

RESPONSE: QECH has been written in full on page 5 line 97. UMIN-CTR has been written in full first (see lines 105-106 on page 5). BACPR has also been written in full first (see lines 110-111 on page 5). All changes are in yellow color. 

REVIEWER: Page 7, line 147: (r)elated data. 

RESPONSE: The comment was not clear. Nevertheless, the () has been inserted around r as shown on line 149; we are not sure if this is what the author meant.

REVIEWER: Page 7, line 169: I suggest adding a quick line of explanation as to why a 20-m long corridor was used for 6MWT, while ATS (2003) recommended a minimum of 25/30 metres. Limitations happen in clinical settings; maybe a short note to close the loop for the benefit of future readers. 

RESPONSE: A statement has been added, justifying why 20m long corridor was used (see lines 168-170)

REVIEWER: Furthermore, please elaborate if participants were required to undergo the 6MWT once or twice. The information is missing. 

RESPONSE: Participants underwent the 6MWT at each time point (baseline, 8-, 12-, 16- and 24-weeks) and each participant had a single session of 6MWT on each day of screening See lines 177-179

REVIEWER: Page 8, lines 184-186: Incomplete statement

RESPONSE: The statement has been completed ‘using free weights in form of weighted sand bags’ (See line 187)

REVIEWER: Page 8, lines 186-187: please further elaborate on how the 1RM testing was performed. 

RESPONSE: A statement has been added to elaborate how the 1RM was conducted. ‘Whilst in sitting position, the researcher handed over the load in the participant’s hand, and they were advised to lift it up to their chest by moving the elbow joint (flexion and extension) without moving the shoulder joint. Participants who were able to lift the same load up for the second time were asked to rest for a minimum of 15 minutes and were allowed to select another increased load to repeat the test until the 1RM was identified.’ (See lines 185-193)

REVIEWER: Page 9, line 195: maybe replace the word guardians with caregivers? Given that the participants are grown-up adults. 

RESPONSE: Guardian has been replaced with caregiver throughout the document (see thoughout the document

Methods in general

REVIEWER: please elaborate on how the data/result corresponding to “healthcare workers’ reported challenges to delivering CR at QECH” were collected. The whole section is missing from the Methods. 

RESPONSE: A statement has been added to the methods section ‘The Physiotherapists involved in CR delivery reported the challenges they encountered in delivering care to the participants, and this reporting was done in writing using a checklist’ (see lines 118-124)

Results: 

REVIEWER: What is the final statistical power with n=60? That differed from the original sample size calculation of n=80. 

RESPONSE: An explanation on why the sample size of n=80 was not reached is explained in lines 123-129. ‘The sample size for this study was calculated to detect a mean difference of 0.6 ml.kg-1.min-1 in the Pred.VO2peak (23) between the groups, at 80% power with a two-tailed significance level set at 0.05. A total of 80 participants, 40 in each arm (1:1), was required. Due to COVID-19 restrictions in the operations of the chest clinic, patient turnover and recruitment was remarkably reduced during the originally proposed study period. The study was subsequently extended by four months to account for these circumstances. The study team managed to contact 73 patients who met the study inclusion criteria and enrolled 60 CHF participants (25% deficit of the actual calculated sample size) over the extended study period’.

REVIEWER: Figure 2 & 3: maybe suggest revising the legends of the graph from “pre-heart rate response” and “post-heart rate response” to “Pre-ET heart rate response” and “Post-ET heart rate response.” I found it confusing initially, uncertain if it refers to pre-post chest step test or pre-post exercise training (CR). 

RESPONSE: The legends have been modified accordingly (pre-heart rate response/pre-rating of perceived exertion and post HR/RPE have all been replaced with pre-ET HR/RPE responses accordingly. See figures 1 and 2)

Discussion

REVIEWER: it would be interesting if the authors could discuss the intermediate long term training effects of this novel hybrid CR, which is currently missing. I read that the improvement of 6MWD in the ET group improved by 103.59+/-48.19 metres (table 4). However, the comparison of the change in 6MWD during the maintenance phase 30.36+/-31.3 and 43.53+/-73.8 metres at 1- and 3-months follow-up respectively. 

RESPONSE: A statement has been added to the discussion section accordingly (See lines 527-536)

Kind regards,

Alice Namanja

---

## [Decision Letter · Decision Letter 1]

9 Jan 2024

Delivering effective, comprehensive, multi-exercise component Cardiac Rehabilitation (CR) for chronic heart failure patients in low resource settings in sub-Saharan Africa: Queen Elizabeth Central Hospital - (QECH-CR) randomised CR study, Malawi.

PONE-D-23-25863R1

Dear Dr. Namanja,

We’re pleased to inform you that your manuscript has been judged scientifically suitable for publication and will be formally accepted for publication once it meets all outstanding technical requirements.

Kind regards,

Yoshihiro Fukumoto

Academic Editor

PLOS ONE

Additional Editor Comments (optional):

Reviewers' comments:

Reviewer's Responses to Questions

**Comments to the Author**

1. If the authors have adequately addressed your comments raised in a previous round of review and you feel that this manuscript is now acceptable for publication, you may indicate that here to bypass the “Comments to the Author” section, enter your conflict of interest statement in the “Confidential to Editor” section, and submit your "Accept" recommendation.

Reviewer #1: All comments have been addressed

2. Is the manuscript technically sound, and do the data support the conclusions?

Reviewer #1: (No Response)

3. Has the statistical analysis been performed appropriately and rigorously? 

Reviewer #1: (No Response)

4. Have the authors made all data underlying the findings in their manuscript fully available?

Reviewer #1: (No Response)

5. Is the manuscript presented in an intelligible fashion and written in standard English?

Reviewer #1: (No Response)

6. Review Comments to the Author

Reviewer #1: (No Response)

7. PLOS authors have the option to publish the peer review history of their article (what does this mean?). If published, this will include your full peer review and any attached files.

Reviewer #1: No
